# Cognitive Considerations in Major Depression: Evaluating the Effects of Pharmacotherapy and ECT on Mood and Executive Control Deficits

**DOI:** 10.3390/brainsci12030350

**Published:** 2022-03-04

**Authors:** Alfredo Spagna, Jason Wang, Isabella Elaine Rosario, Li Zhang, Meidan Zu, Kai Wang, Yanghua Tian

**Affiliations:** 1Department of Psychology, Columbia University in the City of New York, New York, NY 10027, USA; jw3730@columbia.edu (J.W.); ier2108@columbia.edu (I.E.R.); 2Institute for Brain and Spinal Cord, Sorbonne University, 75013 Paris, France; 3Anhui Mental Health Center, Hefei 230022, China; cocozhangli@126.com; 4Department of Psychology and Sleep Medicine, The Second Hospital of Anhui Medical University, Hefei 230601, China; 18355180890@163.com; 5Department of Neurology, The First Hospital of Anhui Medical University, Hefei 230022, China; 6Institute of Artificial Intelligence, Hefei Comprehensive National Science Center, Hefei 230031, China; 7Anhui Province Key Laboratory of Cognition and Neuropsychiatric Disorders, Hefei 230032, China; 8School of Mental Health and Psychological Sciences, Anhui Medical University, Hefei 230032, China; 9Anhui Province Clinical Research Center for Neurological Disease, Hefei 230032, China; 10Department of Neurology, The Second Hospital of Anhui Medical University, Hefei 230601, China

**Keywords:** major depression, attention, electroconvulsive therapy

## Abstract

Deficits in the executive control of attention greatly impact the quality of life of patients diagnosed with major depressive disorder (MDD). However, attentional deficits are often underemphasized in clinical contexts compared with mood-based symptoms, and a comprehensive approach for specifically evaluating and treating them has yet to be developed. The present study evaluates the efficacy of bifrontal electroconvulsive therapy (ECT) combined with drug therapy (DT) in alleviating mood-related symptomatology and executive control deficits in drug-refractory MDD patients and compares these effects with those observed in MDD patients undergoing DT only. The Hamilton Rating Scale for Depression and the Lateralized Attentional Network Test-Revised were administered across two test sessions to assess treatment-related changes in mood-based symptoms and conflict processing, respectively, in patients undergoing ECT + DT (*n* = 23), patients undergoing DT (*n* = 33), and healthy controls (*n* = 40). Although both groups showed an improvement in mood-based symptoms following treatment and a deficit in conflict processing estimated on error rate, a post-treatment reduction of an executive control deficit estimated on RT was solely observed in the ECT + DT patient group. Furthermore, Bayesian correlational analyses confirmed the dissociation of mood-related symptoms and of executive control measures, supporting existing literature proposing that attentional deficits and mood symptoms are independent aspects of MDD. The cognitive profile of MDD includes executive control deficits, and while both treatments improved mood-based symptoms, only ECT + DT exerted an effect on both measures of the executive control deficit. Our findings highlight the importance of considering the improvement in both mood and cognitive deficits when determining the efficacy of therapeutic approaches for MDD.

## 1. Introduction

Affecting over 17 million adults in the U.S. alone [1] and over 300 million individuals worldwide [2], major depressive disorder (MDD) is predicted to be the leading cause of global disease burden by 2030 [2]. This disorder has a debilitating impact on many aspects of daily life, contributing to domestic stress, disability-related absenteeism, and financial insecurity [3]. Far from being considered a homogeneous diagnostic category, MDD symptomatology and severity are widely variable. This poses a great challenge to selecting the most effective course of treatment. According to the DSM-5, the diagnostic criteria of MDD encompass a broad range of symptoms beyond depressed mood, including but not limited to psychomotor agitation, weight loss, impaired concentration, anhedonia, and fatigue [4]. Recognition of the variability of symptom patterns has contributed to the recent classification of symptom-based subtypes, or symptom “fingerprints”, as an approach to more effectively individualize treatment strategies [5,6]. Given the relationship of MDD to poor functional outcomes, increased risk of suicide [2], and a high likelihood of recurrence [7], it is imperative to develop effective treatments and interventions for this disorder.

Cognitive deficits [8] play a critical role in the life of an individual living with depression. However, they are often underemphasized in clinical practice [9] and may not be reliably evaluated using depression rating scales [10]. The range of cognitive deficits associated with MDD is not clearly specified within current diagnostic criteria [11], and there are no standard clinical guidelines for the administration of cognitive assessments in clinical evaluations of MDD. Further, a majority of cognitive instruments used in clinical settings [12] and in clinical trials [13] fail to meet psychometric requirements or are not assessing the specific cognitive domains affected by MDD. The lack of consensus regarding the appropriate assessment of cognitive deficits associated with MDD is a striking issue given their significant impact on psychosocial outcomes and overall quality of life [14]. Additionally, the contributions of various cognitive deficits to both the onset and the maintenance of depression, including the exacerbation of mood-related symptoms and slowed psychomotor processing [15], highlight their clinical relevance and a need to shift away from conceptualizing MDD as a primarily mood-based disorder [9].

Among the cognitive deficits associated with the profile of MDD, attentional deficits have been extensively identified as a key impacted domain and as a vulnerability factor for MDD [16]. Their presence often contributes to a higher tendency to rely on maladaptive emotional regulation strategies to cope with low mood and negative events [17,18]. This aligns with evidence suggesting that attentional deficits are a component of MDD that exist independently of mood symptoms [19], and that the severity of attentional deficits worsens with recurrent depressive episodes [20] and is associated with reduced quality of life and psychosocial functioning [21]. On the other hand, a reduction of attentional deficits following cognitive remediation intervention has been associated with improved psychosocial functioning [22], and a sustained remission of symptoms has been shown to extend as far as 6 months after the conclusion of ECT treatment [23]. Altogether, these findings demonstrate how the presence of attention deficits is not only a vulnerability factor for individuals with depression but also a viable target for indirectly ameliorating their clinical symptomatology.

The executive control of attention [24,25] seems to be particularly relevant to MDD symptomatology since deficits in this component of attention can further the severity of rumination [26] and suicidal ideation [27]. Defined as the mechanism allowing the selection and prioritization of the processing of goal-related information to reach consciousness [24,28,29], this function has been extensively studied using tasks requiring participants to inhibit the effect of distracting information for the purpose of efficient selection of a target stimulus [19,30,31,32], such as the flanker task [33], the Stroop task [34], and the Simon task [35]. MDD status has been linked to longer response times on incongruent trials measured by the Stroop tasks (e.g., [36,37]) and the Attention Network Test (e.g., [38,39]) in adults as well as in adolescents [40,41], suggesting that executive control deficits manifest across different age ranges. Characterizing the extent and variability of executive control deficits in MDD would refine our ability to identify specific symptom fingerprints and optimize treatments.

Pharmacotherapy (DT) and electroconvulsive therapy (ECT) are two treatment options for MDD that can affect attentional functions. Examples of commonly prescribed antidepressant medications include venlafaxine [38], ketamine [42], and sertraline [43]. In a recent study [38], we showed that venlafaxine, an antidepressant pertaining to a class of serotonin–norepinephrine reuptake inhibitors (SNRIs), selectively ameliorates the deficit in the executive control of attention observed in MDD patients. In addition to venlafaxine, escitalopram and sertraline have also been found to ameliorate executive control deficits associated with MDD [44]. On the other hand, MDD patients experiencing treatment-resistant depression (i.e., who exhibit low responses to multiple trials of DT) can be assigned to ECT treatments. Evidence has shown that ECT patients exhibit improvements in executive control functions [45] observed to last anywhere between 6 weeks [46] and 6 months [47] post-treatment. Although this field of research is steadily expanding, there is still a lack of studies that compare the efficacy of DT and ECT in resolving these deficits. We aim to tackle this issue in the present article, as a comparative analysis of these two treatments is critical to inform future therapeutic interventions.

To achieve this goal, we used a computerized attentional test (the Lateralized Attention Network Test-Revised [29]) in a pretest–posttest design to examine changes in both clinical symptoms and measures of executive control in three groups: (1) a group of patients undergoing drug therapy (DT group, *n* = 33), (2) a group of patients undergoing electroconvulsive therapy in combination with drug therapy (ECT + DT group, *n* = 23), and (3) a group of healthy controls serving as the control group (HC group, *n* = 40). We predicted that both the DT and ECT + DT groups would show a greater conflict effect compared with the HC group, indicating the existence of a deficit in the executive control of attention associated with MDD. Further, we hypothesized that clinical symptoms and executive control deficits would be significantly reduced after either DT or ECT + DT treatment, and we sought to compare the efficacy of these treatments.

## 2. Materials and Methods

### 2.1. Participants

Sixty-seven outpatients with MDD were recruited from Anhui Mental Health Center affiliated with Anhui Medical University, China. Diagnosis of MDD was determined by the consensus of two independent psychiatrists using the Structured Clinical Interview for DSM-IV [48]. Eleven patients received a diagnosis of bipolar depression and were therefore excluded from the analysis. Patients with treatment-resistant depression and/or suicidal ideation who were prescribed ECT were recruited to participate in our study as part of the ECT + DT (*n* = 23; 4 males, 19 females). Thirty-three patients entered our DT group (*n* = 33; 8 males, 25 females). See Table 1 for additional demographic, clinical, and treatment information on both patient groups and the HC group. Patients were monitored for dose titration and adverse side effects. The 17-item Hamilton Rating Scale for Depression (HAMD) [49] was used to measure the severity of clinical symptoms and was administered 12–24 h before the beginning of the first testing session (pretest) and between 24 and 72 h from the last testing session (post-test). The time between pre- and postassessments for the DT group was 28 ± 68 days on average. Both the ECT + DT group and the DT group completed the LANT right after completing the pre- and post-HAMD evaluations. Exclusion criteria included the following: history of brain tumor, stroke, or other neurological diseases that could disrupt brain function; presence of psychotic or organic mental disorders; diagnosis of bipolar I disorder; current alcohol or drug dependence; diagnosis of borderline or antisocial personality disorder; current treatment with other psychotropic medications; current or recent pregnancy; a score below 24 in the Mini Mental State Examination; and less than 5 years of schooling.

Forty healthy controls (8 males and 32 females) were recruited from the local area. Participants in the HC group were evaluated by staff psychiatrists, and individuals with a history of neurological, psychiatric, or systemic medical disorders were not included. The time between pre- and postassessments for the HC group was 21 ± 4 days on average. All participants had normal or corrected-to-normal vision and gave written informed consent. The ethical committee of Anhui Medical University approved this study, and the methods and procedures of this study were in accordance with the approved guidelines.

### 2.2. Lateralized Attention Network Test-Revised (LANT-R)

Participants completed a lateralized version of the Attention Network Test-Revised (LANT-R), as implemented in our previous studies [29,50]. The LANT-R consists of a simple computerized task requiring the participant to indicate the direction of an arrow, presented at 6 degrees of the visual field to the right or to the left of a central fixation point, by performing an up or down button press. The presentation of the target was preceded by one of three cue conditions: (1) double cue, (2) spatial cue, and (3) no cue, to manipulate the participants’ attentional orientation to the target. An overview of the task design and presented stimuli is illustrated in Figure 1, and a detailed description of the task can be found in the Appendix A.

### 2.3. ECT Protocol

Patients with treatment-resistant depression underwent modified bifrontal ECT, the standard at Anhui Mental Health Center, using a Thymatron System IV Integrated ECT Instrument (Somatics, Lake Bluff, IL, USA). All patients continued to take antidepressant medication while undergoing ECT treatment (hence, ECT + DT). All the ECT administrations were conducted at Anhui Mental Health Center, with the first three administrations occurring on consecutive days, and the remaining conducted every other day with a break of weekends until patients’ symptoms remitted, defined as a HAMD score lower than 7 (see also [51,52]). The average total duration of the ECT treatment was 14.6 ± 5.8 days. The initial percent energy dial was set based on the age of each participant. If the patient was older than 50 years, the initial percent energy dial setting was set to the patient’s age (for example, 53% for a 53-year-old patient), and if not, the initial percent energy dial was set as the patient’s age minus five (for example, 40% for a 45-year-old patient). Seizure activity was monitored and estimated using electroencephalography, performed concurrently with ECT. If no seizure activity resulted, the percent energy would increase until a therapeutically satisfactory seizure was obtained. During each ECT procedure, the patients were under propofol anesthesia, with succinylcholine and atropine administered to relax muscles and suppress the secretion of glands. Seizure activity was monitored using electroencephalography, and patients continued to take their regular antidepressant medication during ECT treatment.

### 2.4. Data Analysis

Mean response time (RT) and error rate (ER) on the congruent and incongruent conditions LANT-R were first calculated for each subject, with error trials (incorrect and missing responses) and outliers (RTs above three standard deviations) being excluded from the analyses on the RT. The conflict effect, used to measure the executive control of attention, was then estimated on the mean RTs and mean ERs by computing the difference between performances on incongruent and congruent trials. Analysis of the conflict effect estimated on RT and ER was performed according to both frequentist and Bayesian approaches, with a frequentist significance level of α = 0.05 and Bayes factor thresholds of 3 and ⅓ for accepting the alternative and null hypotheses, respectively. The null hypothesis always referred to the absence of a difference. Key conclusions from the analyses were drawn when results from frequentist and Bayes factor analyses were consistent.

The following analyses were then conducted on JASP (JASP Team, 2020): one-way ANOVAs were conducted to examine between group (HC, ECT + DT, DT) differences in Age (in years) and Education (in years) at pretest. A Group (HC, ECT + DT, DT) × Session (Pre, Post) ANOVA was then conducted on HAMD scores to examine whether the ECT and DT groups’ respective treatments were effective in reducing clinical symptoms. We then conducted a Group (HC, ECT + DT, DT) × Session (Pre, Post) × Hemisphere (LH, RH) ANOVA on the conflict effect measures to examine treatment-related changes in the executive control of attention and potential interactions with treatment type and with the side of presentation of the stimuli. Frequentist effect sizes are reported as partial eta squared (η_p_^2^), and Bayesian effect sizes are reported as BF_incl_ (estimated as the relative strength of all models containing a certain factor compared with models not containing the same factor).

Nonparametric planned comparisons (Mann–Whitney U tests, Wilcoxon signed-rank tests) were conducted to analyze the difference between levels of interest in the significant interactions since either some levels of our variables were not normally distributed, or the assumption of equality of variance was not met, which was expected due to the complexity of our experimental design. Nonparametric planned comparisons (Mann–Whitney U tests, Wilcoxon signed-rank tests) were also conducted to compare the length of illness duration between the two patient groups. Medians and the interquartile range (IQR) are reported as the descriptive statistics in the nonparametric analyses. Results from Mann–Whitney U tests and Wilcoxon signed-rank tests are reported as U and W, respectively. BF_10_ values estimate the relative probability of the alternative model compared with the null model. Rank-biserial correlations of the nonparametric post hoc planned comparisons (estimated as the normalized (Z) value from the test divided by the total number of observations) are reported as ȓ. The posterior median effect sizes obtained from corresponding Bayesian nonparametric tests (δ) along with the 95% credible interval (CI) are reported for all planned comparisons.

To further examine the results of the interaction between Group (HC, ECT + DT, DT) × Session (Pre, Post) conducted on the conflict effect (CE) estimated on RT and the heterogeneity of individual measures, exploratory hierarchical cluster analyses based on Ward’s method of minimum variance with a squared Euclidean distance measure were performed on the two patient groups (ECT + DT, DT) to identify patterns of presession CE RT magnitudes and their potential impact on postsession measurements. The size of clusters was based on cluster validity indices and inspection of the dendrogram. Cluster × Session ANOVAs for each patient group were conducted on HAMD scores and the CE estimated on RT to identify potential differences in clinical and attentional measures among the constituent clusters. Nonparametric comparisons (Mann–Whitney U tests) were conducted to analyze within-cluster differences on clinical and attentional measures.

To investigate whether changes in clinical symptomatology were associated with changes in attentional performance, frequentist and Bayesian nonparametric correlation analyses were conducted among the pretest minus post-test differences in HAMD scores and the pretest minus post-test differences in the conflict effects estimated on RTs and ERs. Lastly, in order to investigate whether the length of illness duration and pretreatment illness severity were correlated with pretreatment executive control measures in the ECT + DT and DT groups, frequentist and Bayesian nonparametric correlation analyses were conducted among illness duration (in months), HAMD scores, and pretreatment measures of the mean conflict effects estimated on RTs and ERs. All correlations are reported as Kendall’s tau (τ) and are accompanied by both frequentist *p*-values and Bayesian BF_10_ values.

## 3. Results

### 3.1. Demographic and Clinical Differences across Groups

The one-way ANOVAs conducted separately on the variables Age and Education showed that the factor Group did not reach statistical significance (Age: F < 1; BF_incl_ = 0.10; Education: F(2,93) = 2.01; *p* = 0.14; η_p_^2^ = 0.04; BF_incl_ = 0.49). The Mann–Whitney U test conducted to compare differences in illness duration between the ECT and DT groups did not reach statistical significance (U = 451; *p* = 0.24; ȓ = 0.19; BF_10_ = 0.38; δ = 0.20; 95% CI (−0.29, 0.69)).

We then conducted an ANOVA on HAMD scores with the factors Group (HC, ECT, DT) and Session (pre, post). Here we report results from the significant two-way interaction (F(2,93) = 260.79; *p* < 0.001; η_p_^2^ = 0.85; BF_incl_ = 2.11 × 10^48^), which was one of the comparisons of interest in the present article, while additional results from this ANOVA can be found in the Appendix A. Pairwise comparisons showed evidence for the absence of a difference between pretest and post-test scores for the HC group (W = 198.50; *p* = 0.32; ȓ = 0.22; BF_10_ = 0.25; δ = 0.14; 95% CI (−0.16, 0.44)), whereas there was strong evidence for this difference for both the ECT + DT (W = 276.00; *p* < 0.001; ȓ = 1.00; BF_10_ = 8.53 × 103; δ = 2.56; 95% CI (1.02, 4.33)) and DT (W = 561.00; *p* < 0.001; ȓ = 1.00; BF_10_ = 2.47 × 103; δ = 2.26; 95% CI (1.15, 3.78)) groups. Interestingly, while the DT group showed significantly greater clinical symptoms in the pretest session measurement compared with the ECT + DT group (U = 682.00; *p* < 0.001; ȓ = 0.80; BF_10_ = 204.59; δ = 0.52; 95% CI (0.01, 1.07)), there was no longer a significant difference following the post-test session (U = 460.00; *p* = 0.18; ȓ = 0.21; BF_10_ = 0.41; δ = 0.20; 95% CI (−0.29, 0.68)), indicating that the two treatments reduced the clinical symptoms to similar levels. However, the difference in the HAMD scores between the HC group and the ECT + DT group (U = 761.50; *p* < 0.001; ȓ = 0.66; BF_10_ = 121.55; δ = 0.96; 95% CI (0.42, 1.50)) as well as the DT group (U = 1227.50; *p* < 0.001; ȓ = 0.86; BF_10_ = 1.96 × 103; δ = 1.30; 95% CI (0.78, 1.84)) was still significant post-test, indicating the remission of the symptomatology after treatment was only partial (see Figure 2).

### 3.2. Treatment Effects on Executive Control Measures Estimated on RT

Appendix A provides summary statistics for RTs and ERs on congruent and incongruent trials split by the field of stimulus presentation for each group, while Table 2 and Table 3 summarize results from the ANOVAs conducted on the conflict effect (CE) estimated on RT, while Table 4 and Table 5 summarize results from the ANOVAs conducted on the conflict effect (CE) estimated on ERs. Here we report results from the significant two-way interaction Group × Session estimated on the conflict effect (CE) on RT (F(2,93) = 4.52; *p* < 0.05; η_p_^2^ = 0.09; BF_incl_ = 85.56), which was one of the comparisons of interest in the present article. Additional results from this ANOVA are discussed in the Appendix A.

In the pretest session, the CE estimated on RT was greater in the ECT + DT group compared with the HC group (U = 628; *p* < 0.05; ȓ = 0.37; BF_10_ = 1.48; δ = 0.46; 95% CI (−0.03, 0.96)). The difference between the DT group and the HC group did not reach statistical significance (U = 759; *p* = −0.28; ȓ = 0.15; BF_10_ = 0.37; δ = 0.19; 95% CI (−0.24, 0.63)). Furthermore, there was strong evidence for the absence of a difference between the ECT + DT and DT groups (U = 331; *p* = 0.424; ȓ = −0.13; BF_10_ = 0.320; δ = −0.13; 95% CI (−0.64, 0.34)).

In the post-test session, the difference between the HC group and the ECT + DT group was no longer significant (U = 358; *p* = 0.15; ȓ = 0.22; BF_10_ = 0.74; δ = 0.31; 95% CI (−0.80, 0.17)). There was strong evidence for the absence of a difference between the HC group and the DT group (U = 597; *p* = 0.49; ȓ = 0.10; BF_10_ = 0.27; δ = 0.08; 95% CI (−0.36, 0.51)). Lastly, the difference between the ECT + DT group and the DT group did not reach statistical significance (U = 421.50; *p* = 0.49; ȓ = 0.11; BF_10_ = 0.42; δ = −0.23; 95% CI (−0.73, 0.24)).

As per the within-group comparisons, there was no significant difference between the pretest session and post-test session CE estimated on RT for both the HC (W = 493; *p* = 0.267; ȓ = 0.20; BF_10_ = 0.52; δ = 0.233; 95% CI (−0.079, 0.53)) and DT groups (W = 355.5; *p* = 0.18; ȓ = 0.27; BF_10_ = 0.66; δ = 0.27; 95% CI (−0.07, 0.60)), while there was strong evidence for a difference between the pretest and post-test session in the ECT + DT group (W = 250; *p* < 0.001; ȓ = 0.81; BF_10_= 73.08; δ = 0.90; 95% CI (0.40, 1.45)).

### 3.3. Exploratory Cluster Analysis Conducted Separately for the ECT and DT Groups on RT CE

Results from the exploratory hierarchical clustering analysis according to pretest RT CE measures are reported on each patient group. The clustering analysis conducted on the ECT + DT group resulted in two different clusters (cluster 1: *n* = 7; cluster 2: *n* = 16), while the clustering analysis conducted on the DT group resulted in three different clusters (cluster 1: *n* = 13; cluster 2: *n* = 14; cluster 3: *n* = 6). Appendix A provides a summary of the agglomeration coefficients used to select the optimal clustering method, while Appendix A summarize the descriptive statistics of each cluster.

For both the ECT + DT and DT cluster analyses conducted on HAMD scores, we found evidence for the absence of a Session × Cluster interaction (ECT + DT: F(1,30) = 0.10; *p* = 0.76; η_p_^2^ = 0.01; BF_incl_ = 0.29; DT: F(1,30) = 0.49; *p* = 0.62; η_p_^2^ = 0.03; BF_incl_ = 0.28), adding more evidence that mood symptoms and attentional deficits are not associated.

The ANOVA performed on the CE RT in the ECT + DT clusters showed strong evidence for the Session × Cluster interaction (F(1,21) = 23.24; *p* < 0.001; η_p_^2^ = 0.53; BF_incl_ = 217.60). Post hoc comparisons showed that both Cluster 1 (W = 28.00; *p* = 0.02; ȓ = 1.00; BF_10_ = 16.27; δ = 2.28 (0.34, 7.77)) and Cluster 2 (W = 111.00; *p* = 0.03; ȓ = 0.63; BF_10_ = 4.82; δ = 0.63 (0.11, 1.2)) showed strong evidence for the reduction of CE in the post-test session. Lastly, while there was strong evidence for the CE of Cluster 1 being greater than that of Cluster 2 (U = 112.00; *p* < 0.001; ȓ = 1.00; BF_10_ = 9.1; δ = 1.15 (0.20, 2.19)) at the pretest session, the difference was no longer significant at the post-test session (U = 112.00; *p* = 0.62; ȓ = 0.14; BF_10_ = 0.43; δ = 0.15 (−0.57, 0.94)).

The ANOVA performed on the CE RT in the DT group showed strong evidence for the Session × Cluster interaction (F(2,30) = 14.40; *p* < 0.001; η_p_^2^ = 0.49; BF_incl_ = 727.59). Post hoc comparisons showed that within Cluster 1, there was strong evidence for a difference in the pretest (W = 6.00; *p* = 0.003; ȓ = −0.87; BF_10_ = 21.66; δ = −1.03 (−1.82, −0.31)), with patients having, surprisingly, a greater magnitude CE in the post-test session.

Within Cluster 2, there was strong evidence for a reduction in the CE from the pretest to the post-test session (W = 91.00; *p* = 0.017; ȓ = 0.73; BF_10_ = 9.30; δ = 0.75 (0.18, 1.41)).

Within Cluster 3, there was inconclusive evidence for a difference between the pretest and the post-test (W = 19.00; *p* = 0.09; ȓ = 0.81; BF_10_ = 1.94; δ = 0.76 (−0.10, 1.837)).

For the between-group comparisons, at the pretest the CE of Cluster 1 was significantly lower than that of Cluster 2 (U = 0; *p* < 0.001; ȓ = −1.00; BF_10_ = 47.10; δ = −1.39 (−2.33, −0.48)) and Cluster 3 (U = 0; *p* < 0.001; ȓ = −1.00; BF10 = 4.15; δ = −1.08 (−2.23, −0.05)), while at the post-test session Cluster 1 did not significantly differ from Cluster 2 (U = 97.00; *p* = 0.79; ȓ = 0.07; BF_10_ = 0.37; δ = 0.003 (−0.65, 0.68)) or Cluster 3 (U = 26.00; *p* = 0.27; ȓ = −0.33; BF10 = 0.71; δ = −0.39 (−1.33, 0.39)). The CE of Cluster 2 was significantly lower than that of Cluster 3 (U = 0; *p* < 0.001; ȓ = −1.00; BF_10_ = 5.07; δ = −1.10 (−2.34, −0.08)) at the pretest session, while there was no significant difference at the post-test session (U = 32.00; *p* = 0.43; ȓ = −0.24 BF_10_ = 0.49; δ = −0.21 (−1.08, 0.53)).

Figure 3 depicts the results from the Group × Session ANOVA on the CE estimated on RT (panel 1: HC vs. ECT + DT; panel 2: HC vs. DT; panel 3: ECT + DT vs. DT) and the exploratory hierarchical cluster analyses for the ECT + DT group (panel 4) and DT group (panel 5).

### 3.4. Treatment Effects on Executive Control Measures Estimated on ER

The ANOVA conducted on the conflict effect estimated on error rates (ER) with the factors Group (HC, ECT + DT, DT), Session (pre, post), and Hemisphere (LH, RH) showed strong evidence for the main effect of Group (F(2,93) = 7.56; *p* < 0.001; η_p_^2^ = 0.14; BF_incl_ = 40.97). The overall CE (ER) of the ECT + DT group (median (IQR): 2.08 (3.82)%) was greater than that of the HC group (median (IQR): 0.52 (2.13)%; U = 624; *p* = 0.02; ȓ = 0.36; BF_10_ = 1.72; δ = 0.50 (0.002, 1.01)). There was strong evidence for the overall CE (ER) of the DT group (median (IQR): 4.52 (5.56)%) being significantly greater than that of the HC group (U = 1104; *p* < 0.001; ȓ = 0.67; BF_10_ = 361.15; δ = 1.04 (0.53, 1.55)). The overall CE (ER) of the DT group was also significantly greater than that of the ECT group (U = 537; *p* = 0.009; ȓ = 0.42; BF_10_ = 2.24; δ = 0.52 (0.01, 1.07)). However, the Group × Session interaction failed to reach significance according to the frequentist approach (F(2,93) = 2.54; *p* = 0.08; η_p_^2^ = 0.05), while Bayesian analysis showed strong evidence for its presence (BF_incl_ = 5.18) (see Figure 4). Follow-up analyses showed that the difference between the pretest and post-test CE did not reach statistical significance for the HC group (W = 279.5; *p* = 0.08; ȓ = −0.32; BF_10_ = 0.30; δ = −0.16 (−0.47, 0.14)), for the ECT + DT group (W = 155; *p* = 0.36; ȓ = 0.23; BF_10_ = 0.37; δ = 0.20 (−0.19, 0.60)), and for the DT group (W = 375; *p* = 0.09; ȓ = 0.34; BF_10_ = 1.10; δ = 0.31 (−0.02, 0.65)).

For the between-group differences, the CE of the ECT + DT group was greater than that of the HC group at the pretest session (U = 651; *p* = 0.007; ȓ = 0.42; BF_10_ = 6.29; δ = 0.64 (0.12, 1.16)) but no longer significantly differed at the post-test session (U = 536; *p* = 0.28; ȓ = 0.17; BF_10_ = 0.42; δ = 0.23 (−0.23, 0.73)). The CE of the DT group was greater than that of the HC group at both the pretest (U = 1107; *p* < 0.001; ȓ = 0.68; BF_10_ = 443.59; δ = 1.04 (0.54, 1.56)) and post-test sessions (U = 914; *p* 0.005; ȓ = 0.39; BF_10_ = 4.35; δ = 0.53 (0.08, 1.00)). The CE of the DT group was greater than that of the ECT + DT group at the pretest session (U = 527; *p* = 0.01; ȓ = 0.39; BF_10_ = 2.28; δ = 0.52 (0.02, 1.06)) but no longer significantly differed at the post-test session (U = 475; *p* = 0.11; ȓ = 0.25; BF_10_ = 0.63; δ = 0.32 (−0.18, 0.86)).

### 3.5. Exploratory Correlation Analyses between Executive Control Function, Clinical Symptoms, and Illness Duration

A summary of the results from the correlation analyses can be found in Table 6. Briefly, analyses using the frequentist approach showed that the correlation between illness duration or clinical symptomatology pretreatment and attentional performance did not reach statistical significance. Bayesian correlation analyses among pretreatment executive control measures, clinical symptomatology, and illness duration showed evidence for the absence of correlations in the DT group, while results for the ECT + DT group were at or around the evidential threshold for the absence of a correlation. Correlation analyses conducted separately for each group on changes (pre minus post) in HAMD scores and changes (pre minus post) in the CE estimated on RT and ER did not reach statistical significance for all measures, and Bayesian analysis provided either evidence against or inconclusive evidence for the presence of a correlation between changes in clinical symptoms and changes in the CE across all measures.

## 4. Discussion

In the present study, we find evidence of a behavioral deficit in the executive control of attention associated with major depression and demonstrate the efficacy of both electroconvulsive therapy + drug therapy and drug therapy in reducing clinical symptoms as well as attentional deficit. We also find conclusive evidence for a dissociation between depression severity and executive control deficit, making a case for considering executive control deficits a distinct symptom with clinical relevance.

We found evidence that both ECT + DT and DT significantly reduce clinical symptomatology, but post-treatment scores for each group did not reach levels comparable to healthy controls. In our results, although both the ECT + DT and DT groups showed significantly lower HAMD scores post-treatment, group scores were at or around the cut-off for clinical diagnosis, indicating that core depression symptoms were still observable in at least some of the patients. The HAMD scale ranges between 0 (no symptoms) and 50 (maximum), with scores below 7 considered to be subclinical [53], and it ranks severity by reducing numerous dimensions of the disorder to a single metric. Furthermore, the differential contributions of each item to the final score and the nonuniformity of rating scales across items can limit the consistency of these measures in detecting disorder severity across individuals [54]. The partial reduction of the HAMD scores post-treatment observed in this study may serve as evidence for the limitations of relying only on HAMD scores to classify remission, since it is in line with previous evidence showing how the remission of symptoms after either ECT (e.g., [55]) or DT (e.g., [38,56]) treatment may not always reach subthreshold levels.

We also note that long-term clinical effects of these treatments are beyond the scope of this study. While longitudinal studies independently examining the long-term cognitive effects of either ECT [46,57,58] or DT [59,60,61,62] exist, an investigation of the comparative long-term efficacy between ECT and DT treatments is still needed. Furthermore, we observed an unusual pattern with the ECT + DT group scoring as less severely depressed than the DT sample on the HAMD scores pretest. This pattern was not due to a difference in the illness duration between the two groups or in the number of patients experiencing a first episode of MDD vs. relapsed patients, or differences in dosage of the drug treatment.

We found evidence that executive control deficits exist within the cognitive profile of major depression, and that they are independent of depression severity. In this study, we estimated the flanker conflict effect [33] from the Attention Network Test [28,29] to measure executive control of attention and identified a deficit in this function associated with major depression. Specifically, patients in the ECT + DT group reported a significantly greater conflict effect compared with healthy controls in the pretreatment session. This result is in line with previous evidence showing that executive control deficits are part of the cognitive profile of major depression [19,32,63,64]. For instance, in our previous study [38] we administered the Attention Network Test to a group of patients with major depression, finding abnormal executive control performance estimated on response time and no association between pretreatment executive control measures and core depression symptoms. Prior studies have suggested that the lack of a significant association between attention deficits and mood-related symptoms may indicate the independence of these two phenomena [19,65]. Here, results from both frequentist and Bayesian analyses confirmed the absence of any association between executive control deficits and clinical symptoms across most of our measurements, suggesting that although these two phenomena can manifest alongside one another, they are two independent aspects of this disorder.

We found strong evidence that combining electroconvulsive therapy with pharmacotherapy significantly reduced the executive control deficit post-treatment. As shown in Figure 3, a post-treatment reduction of the conflict effect on response time was observed in the ECT + DT group, while a similar pattern was observed for only one out of three clusters in the DT group. It is important to keep in mind that cluster analyses are to be considered exploratory, and readers should exercise caution when drawing any interpretation from these results. This result may suggest an advantage in favor of the electroconvulsive treatment in addressing the executive control deficit associated with major depression. This result is in line with previous evidence showing a significant improvement post-ECT treatment in both attention and executive control [46,47], and shows some preliminary evidence that the addition of ECT to pharmacological treatment may have an advantage in reducing the executive control deficit over administering DT alone. Overall, the specificity of both treatments in reducing conflict effect deficit suggests that they effectively modulate the executive control of attention in addition to improving mood-related symptoms.

We found evidence for right hemisphere superiority in the executive control of attention and a partial normalization of this effect after treatment. By using a lateralized version of the Attentional Network Test [29,50], in which stimuli were presented in the periphery of the visual field (about 6 degrees of the visual field to the left or right of the central fixation cross) [66], we were able to confirm the advantage of the right hemisphere in conflict resolution (results presented in the Appendix A). Specifically, we showed that the flanker conflict effect estimated on response time was significantly reduced for targets presented in the left visual field (processed in the right hemisphere) compared with targets presented in the right visual field (processed in the left hemisphere), a result due to faster responses to incongruent trials presented in the left visual field. This result is in line with previous evidence using a similar paradigm [29,67], adding to the extensive literature regarding the right hemispheric advantage for attention observed at the behavioral [68,69] and at the neural level (e.g., [70,71]). Interestingly, the right-hemisphere superiority effect interacted with the factor session in the analyses conducted on error rates, which may be due to normalizing effects found in another recent study that used ECT as a treatment for depression [46], even though our results did not show a differential effect based on treatment group. Although beyond the scope of the current study, we observed a reduction in the conflict effect for targets presented in the right visual field that was not observed for contralateral targets, potentially due to a reduction in the conflict effect post-treatment, which also warrants future investigation.

A few methodological aspects of our study are worth considering as limitations. First, group assignment was based on psychiatrist evaluation of severity, efficacy of pharmacological intervention, and acute suicidal risk. No patients underwent ECT treatment only, and the determination of ECT + DT or DT for patients’ clinical treatment was unrelated to the scope of the present study. As ECT + DT patients were considered treatment resistant, it is important to note that while the average illness duration of the ECT + DT group was higher in magnitude (though not statistically significant) compared with the DT group, the average HAMD score was significantly lower in the ECT + DT group compared with the DT group. A diagnosis of treatment-resistant depression has to undergo multiple considerations, and hence, having a lower HAMD score does not necessarily indicate absence of treatment resistance or of suicidal ideation. Moreover, the research team only has access to HAMD scores upon enrollment in the study and not at the onset of depression for either group, which prevents the establishment of patients’ clinical status before treatment and the extent of symptom reduction in drug-refractory patients.

An additional confounding effect can be attributed to our assignment procedure to either the ECT + DT or DT group because ECT treatment was offered only to patients that were treatment resistant to pharmacological intervention, a characteristic that is not shared with the DT group. Although this assignment procedure is dictated by ethical reasons, it remains important to consider how the presence of this difference in the two groups might affect our conclusions. While the sample size of the present study may present some limitations to the generalizability of the present study’s findings, our research group has published a fairly high number of articles investigating behavioral, cognitive, and neural markers of attention in a variety of clinical populations (e.g., [38,51]). Based on our previous experience, we believe that these numbers were sufficient to observe our predicted effects on attentional decreases post-treatment, which is why we did not conduct a priori power analysis. However, it is important to notice that our cluster analyses are underpowered due to the sample sizes, which is why results must be considered only exploratory.

Variability in the efficacy of ECT treatment according to electrode placement and other methodological differences are key factors relevant to the interpretation of the present findings. The bifrontal electrode placement and ECT session frequency employed in this study was carried out according to the treatment standards of the Second Hospital of Anhui Medical University. We recognize that some studies have shown right unilateral ECT to have greater efficacy than bifrontal ECT [72,73], and that there is a recent trend attempting to reduce total treatment duration (e.g., ultrabrief unilateral ECT) [73]. However, the evidence for this difference in efficacy is mixed [74], and it is recognized that differences between individuals (such as head size) may allow bifrontal placement to be better at avoiding brain regions associated with detrimental memory effects [51]. While highly relevant for future investigations, comparing the efficacy of bifrontal vs. unilateral treatment was beyond the scope of the present study. Altogether, we believe that these limiting factors do not invalidate our main claims regarding the existence of executive control deficits associated with MDD and the reduction of this deficit following ECT + DT treatment.

## 5. Conclusions

In conclusion, executive control deficits are another key aspect of MDD and may be relevant for establishing individualized symptom fingerprints. We found evidence that executive control deficits and clinical symptoms are two independent aspects of major depression, and that both ECT + DT and DT are capable of effectively reducing these aspects of the disorder, although the mechanisms underlying these effects are still unclear. A thorough evaluation of the executive control deficit in patients with major depression is critical to refine our diagnostic criteria and to identify symptom fingerprints that could help individualize treatment strategies [5,6]. Basing interventions on the presence and extent of executive control deficits in each patient could better ameliorate their depressive symptoms, increase attentional flexibility and resistance to distraction [70], help shift attention away from self-referential thoughts [18], and give greater control on their financial and organizational capabilities [75], potentially improving their quality of life.

## Figures and Tables

**Figure 1 brainsci-12-00350-f001:**
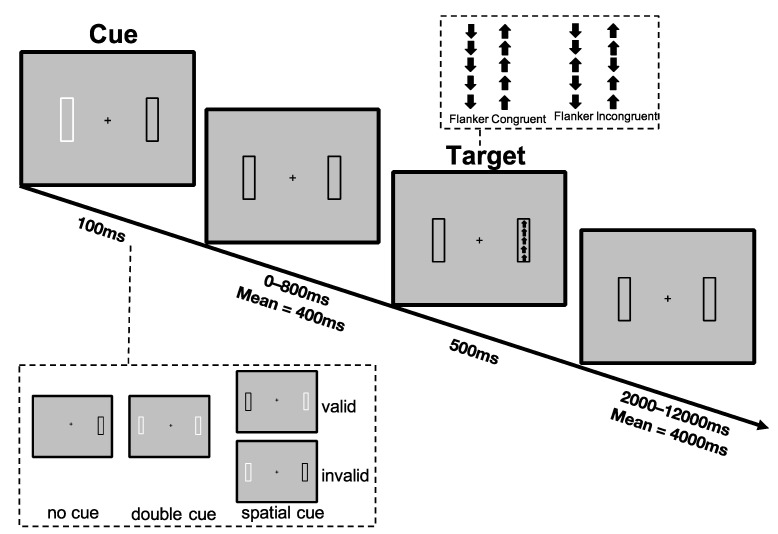
Sequence of events in an invalid cue/congruent trial as implemented by the Lateralized Attention Network Test-Revised (LANT-R) [29,44].

**Figure 2 brainsci-12-00350-f002:**
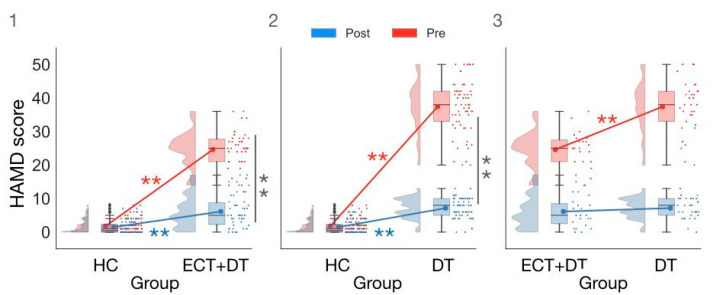
Comparison of HAMD scores in (**1**) HC vs. ECT + DT group; (**2**) HC vs. DT group; (**3**) ECT + DT vs. DT group. ** *p* < 0.001.

**Figure 3 brainsci-12-00350-f003:**
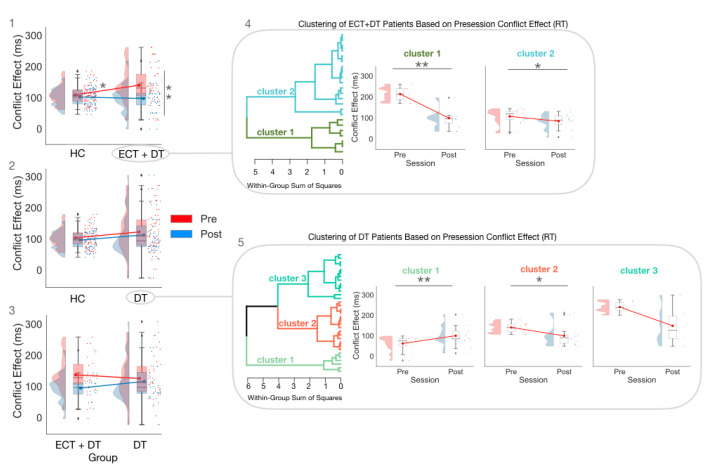
The Group × Session ANOVAs estimated on the CE (RT), with panel (**1**) showing the comparison between HC and ECT + DT, panel (**2**) showing the comparison HC vs. DT, and panel (**3**) showing the comparison between ECT + DT and DT, and subsequent Session × Cluster ANOVAs conducted separately for the ECT + DT (panel (**4**)) and DT (panel (**5**)) clusters. * *p* < 0.05; ** *p* < 0.01. ♦ indicates outlier.

**Figure 4 brainsci-12-00350-f004:**
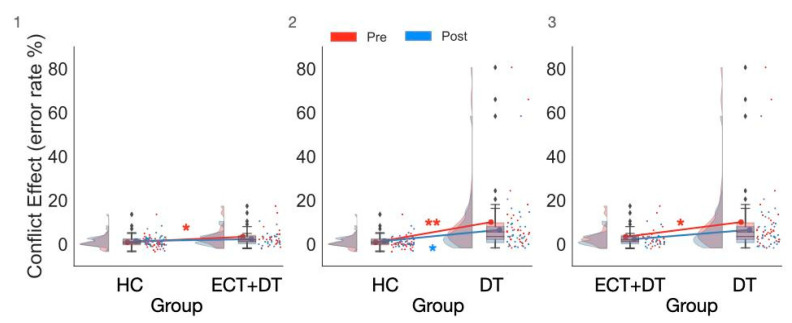
The significant Group × Conflict interaction observed in the Bayesian ANOVA on error rate showed that both the ECT + DT and DT groups made significantly more errors compared with the HC group at pretest, while this difference was significant at post-test only in the DT group. (**1**) Comparison between the HC and ECT + DT groups, (**2**) comparison between the HC and DT groups, (**3**) comparison between the ECT + DT and DT groups. * *p* < 0.01, ** *p* < 0.001. ♦ indicates outlier.

**Table 1 brainsci-12-00350-t001:** Separate demographic and clinical characteristics for each group. Measures of central tendency (median and mean) and of dispersion (IQR and SD) are reported for each variable.

	HC		ECT		DT	
	Mean ± SD	Median (IQR)	Mean ± SD	Median (IQR)	Mean ± SD	Median (IQR)
**Age (years)**	34.55 ± 11.80	30.50 (3.25)	34.39 ± 10.86	31.00 (18)	34.22 ± 11.57	33.00 (19.00)
**Education (years)**	13.53 ± 3.82	15.00 (5.00)	11.87 ± 4.16	14.00 (7.00)	12.06 ± 3.31	11.00 (7.00)
**HAMD Pre**	1.80 ± 1.96	1.00 (2.25)	24.61 ± 5.56	25.00 (6.50)	37.39 ± 7.80	38.00 (9.00)
**HAMD Post**	1.48 ± 1.96	0.50 (2.25)	6.13 ± 4.79	5.00 (6.00)	7.15 ± 3.00	8.00 (5.00)
**Illness Duration (months)**	N/A	N/A	55.91 ± 93.32	12.00 (45.00)	62.09 ± 68.08	36.00 (102.00)
**Relapse**	N/A	N/A	First episode *n* = 12	Relapse *n* = 11	First episode *n* = 12	Relapse *n* = 21
**Gender**	Male *n* = 8	Female *n* = 32	Male *n* = 4	Female *n* = 19	Male *n* = 8	Female *n* = 25

Note: **HAMD** = Hamilton Rating Scale for Depression. **HC** = healthy control participants; **ECT** = patients undergoing electroconvulsive therapy; **DT** = patients undergoing pharmacotherapy; **HAMD Pre** = HAMD score measured at the pretest session; **HAMD Post** = HAMD score measured at the post-test session. IQR = inter-quartile range. In the DT group, venlafaxine = 3 individuals (median dose/day (IQR): 150 (37.50) mg mg); paroxetine = 7 individuals (median dose/day (IQR): 30 (15) mg); sertraline = 6 individuals (median dose/day (IQR): 62.5 (25) mg); duloxetine = 14 individuals (median dose/day (IQR): 60 (20) mg); fluoxetine = 1 individual (median dose/day (IQR): 20 (0) mg); citalopram = 2 individuals (median dose/day (IQR): 12.50 (2.5) mg). In the ECT group, venlafaxine = 3 individuals (median dose/day (IQR): 225 (25) mg); paroxetine = 10 individuals (median dose/day (IQR): 40 (7.5) mg); sertraline = 2 individuals (median dose/day (IQR): 100 (0) mg); duloxetine = 8 individuals (median dose/day (IQR): 60 (5.0) mg).

**Table 2 brainsci-12-00350-t002:** Frequentist and Bayesian results from the ANOVA performed on the conflict effect estimated on response time.

	F	*p*	η_p_^2^	BF_incl_
**Group**	0.98	0.38	0.02	0.21
**Session**	20.52	<0.001	0.18	64,823.21
**Hemisphere**	6.40	0.01	0.06	1.05
**Group × Session**	4.52	0.01	0.09	85.56
**Group × Hemisphere**	1.24	0.29	0.03	0.10
**Session × Hemisphere**	2.16	0.15	0.02	0.29
**Group × Session × Hemisphere**	0.45	0.64	0.01	0.12

**Table 3 brainsci-12-00350-t003:** Descriptive statistics for results from the *Group* by *Session* ANOVA conducted on the conflict effect estimated on response time (ms).

	HC		ECT		DT	
	Mean ± SD	Median (IQR)	Mean ± SD	Median (IQR)	Mean ± SD	Median (IQR)
**Pre**	109.56 ± 30.44	106.75 (36.38)	140.06 ± 61.03	131.00 (63.75)	128.26 ± 70.96	115.00 (81.00)
**Post**	101.45 ± 23.59	103 (28)	88.87 ± 39.22	92.5 (44.5)	108.96 ± 60.70	95 (48)

**Table 4 brainsci-12-00350-t004:** Frequentist and Bayesian results from the ANOVA performed on the conflict effect estimated on error rate (%).

	F	*p*	η_p_^2^	BF_incl_
**Group**	7.56	<0.001	0.14	40.91
**Session**	2.85	0.10	0.03	1.25
**Hemisphere**	2.85	0.80	6.90 × 10^−4^	0.11
**Group × Session**	2.54	0.08	0.05	5.18
**Group × Hemisphere**	2.54	0.91	0.002	0.06
**Session × Hemisphere**	4.90	0.03	0.05	0.55
**Group × Session × Hemisphere**	1.53	0.22	0.03	0.17

**Table 5 brainsci-12-00350-t005:** Descriptive statistics corresponding to the *Group* by *Session* ANOVA conducted on the conflict effect estimated on error rate (%).

	HC		ECT		DT	
	Mean ± SD	Median (IQR)	Mean ± SD	Median (IQR)	Mean ± SD	Median (IQR)
**Pre**	0.88 ± 2.81	0.35 (2.43)	3.34 ± 4.60	2.09 (3.13)	10.03 ± 17.32	5.56 (7.64)
**Post**	1.41 ± 2.21	1.04 (2.43)	2.27 ± 2.98	1.39 (3.30)	6.52 ± 7.29	3.48 (7.29)

**Table 6 brainsci-12-00350-t006:** Coefficients estimated using frequentist and Bayesian nonparametric (Kendall’s tau) and Bayesian correlation analyses among illness duration, clinical symptomatology, and attention performance.

			HC			ECT			DT	
Correlation		τ	*p*	BF_10_	τ	*p*	BF_10_	τ	*p*	BF_10_
**Illness Duration**	**HAMD Pre**	-	-	-	−0.05	0.75	0.28 ^Ϯ^	−0.03	0.83	0.23 ^Ϯ^
**Illness Duration**	**RT Pre CE**	-	-	-	0.11	0.49	0.34	−0.6	0.61	0.26 ^Ϯ^
**Illness Duration**	**ER Pre CE**	-	-	-	−0.05	0.75	0.28 ^Ϯ^	0.002	0.99	0.23 ^Ϯ^
**HAMD Pre**	**RT Pre CE**	-	-	-	0.05	0.75	0.28 ^Ϯ^	−0.07	0.57	0.27 ^Ϯ^
**HAMD Pre**	**ER Pre CE**	-	-	-	0.16	0.29	0.47	0.10	0.41	0.32 ^Ϯ^
**HAMD Diff**	**RT CE Diff**	0.08	0.50	0.27 ^Ϯ^	0.18	0.24	0.53	−0.11	0.38	0.33 ^Ϯ^
**HAMD Diff**	**ER CE Diff**	0.04	0.75	0.22 ^Ϯ^	−0.05	0.75	0.28 ^Ϯ^	0.13	0.29	0.39

Note: **τ** = Kendall’s tau; **HC** = healthy control participants; **ECT** = patients undergoing electroconvulsive therapy; **DT** = patients undergoing pharmacotherapy; **Illness Duration** = length of disease (since first diagnosis) measured in months; **HAMD Pre** = HAMD score measured at the pretest session; **RT Pre CE** = conflict effect estimated on response time (ms) in pretest session; **RT Post CE** = conflict effect estimated on response time (ms) in post-test session; **ER Pre CE** = conflict effect estimated on error rate (%) in pretest sessions; **ER CE Post** = conflict effect estimated on error rate (%) in post-test sessions; ^Ϯ^ BF10 < 0.33.

## Data Availability

For IRB-related reasons, anonymized data cannot be made available to the public.

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
