# Peer review of "Cognitive Considerations in Major Depression: Evaluating the Effects of Pharmacotherapy and ECT on Mood and Executive Control Deficits"

_brainsci, 2022, doi:10.3390/brainsci12030350_

Round 1

Reviewer 1 Report

This is an interesting and relevant study. The methodology is sound, and the statistical analyses are thorough. The paper is well written. However, there are some concerns, mostly related to shortcomings of the Discussion section, that should be addressed: 

  1. This is not really a comparison of ECt versus medication on the effect of depression or cognitive function, as the ECT patients are also medicated. This aspect should be addressed in the Discussion section. Moreover, the Introduction section describes different cognitive effects of different types of medication. This topic is ignored in the Discussion section, although it is highly relevant for the results of this study, as all the patients are medicated and with different types of drugs.
  2. Previous studies have found different symptomatic and cognitive effects of ECT for different types of electrode placement. The current studies employs bifrontal placement. A discussion of the benefits and shortcomings of this method compared to others with respect to symptom levels and cognitive function should be added. 
  3. Some studies have found different outcomes of ECT for females compared to males. Possible gender effects on symptom level and cognitive function could be controlled for statistically, or at least mentioned in the Discussion section as a limitation if it is not done. 
  4. There is a large number of statistical tests being performed on a relatively small sample. Did the authors do any power calculations beforehand? Possibly, the weakest statistically significant results would not withstand a Bonferroni correction. The authors should either perform such correction for multiple testing, or address the absence of such as a limitation. 
  5. Related to all of these concerns: There is no Limitation section in the Discussion part. In fact, there is little discussion of the possible shortcomings of this study at all. Therefore, some of the conclusions drawn are premature. 

Author Response

We would like to begin by thanking the Editor and the four reviewers for their insightful comments and suggestions. Overall, we believe that information we have now added to the manuscript has significantly improved it, especially regarding how patients were assigned to each group. We hope that the reviewers will find our work to be detailed and accurate. We have put our best effort to add as much information as needed, while at the same time shortening the manuscript.

Below we provide a point by point response to each Reviewers’ points, that are now numbered.  We rest at your availability in case additional information is needed.

Reviewer 1

This is an interesting and relevant study. The methodology is sound, and the statistical analyses are thorough. The paper is well written. However, there are some concerns, mostly related to shortcomings of the Discussion section, that should be addressed:

We thank the reviewer for the kind words. We have now modified the manuscript to address those comments.

R1.1. This is not really a comparison of ECT versus medication on the effect of depression or cognitive function, as the ECT patients are also medicated. This aspect should be addressed in the Discussion section. Moreover, the Introduction section describes different cognitive effects of different types of medication. This topic is ignored in the Discussion section, although it is highly relevant for the results of this study, as all the patients are medicated and with different types of drugs.

We agree with the reviewer’s comment, and we have now modified the manuscript accordingly.

First, we have renamed out group as “ECT+DT” instead of ECT, to clarify that some of our participants underwent both treatments in conjunction. This change is reflected both in the main text, in the figures and in the figure captions.

We have slightly modified the discussion to further clarify this aspect (as shown below):

“We found strong evidence that combining electroconvulsive therapy with pharmacotherapy significantly reduced the executive control deficit post-treatment. As shown in Figure 3, a post-treatment reduction of the conflict effect on response time was observed in the ECT+DT group, while a similar pattern was observed for only one out of three clusters in the DT group.”

We also added in the Limitation section a caveat about a potential confounding effect of our assignment procedure to either the ECT+DT or DT group, as shown below

An additional confounding effect can be attributed to our assignment procedure to either the ECT+DT or DT group because ECT treatment was offered only to patients that were treatment-resistant to pharmacological intervention, a characteristic that is not shared with the DT group. Although this assignment procedure is dictated by ethical reasons, it remains important to still consider how the presence of this difference in the two groups might affect our conclusions. While the sample size of the present study may present some limitations to the generalizability of the present study’s findings, our research group has published a fairly high number of articles investigating behavioral, cognitive, and neural markers of attention in a variety of clinical populations e.g., [38, 51]. Based on our previous experience, we believe that these numbers were sufficient to observe our predicted effects on attentional decreases post-treatment, which is why we did not conduct a priori power analysis. Yet, it is important to notice that our cluster analyses are underpowered due to the sample sizes, which is why results must be considered only exploratory.

R1.2. Previous studies have found different symptomatic and cognitive effects of ECT for different types of electrode placement. The current studies employs bifrontal placement. A discussion of the benefits and shortcomings of this method compared to others with respect to symptom levels and cognitive function should be added.

We have now discussed this point in the Discussion section accordingly, as shown below and also on page (XYZ) of the manuscript.

Variability in the efficacy of ECT treatment according to electrode placement and other methodological differences are key factors relevant to the interpretation of the present findings. The bifrontal electrode placement and ECT session frequency employed in this study was carried out according to the treatment standards of the Second Hospital of Anhui Medical University. We recognize that some studies have shown right unilateral ECT to have greater efficacy than bifrontal ECT [73], and that there is a recent trend attempting to reduce total treatment duration (e.g., ultrabrief unilateral ECT) [74](Shafi et al., 2018). However, the evidence for this difference in efficacy is mixed [75], and it is recognized that differences between individuals (such as head size) may allow bifrontal placement to be better at avoiding brain regions associated with detrimental memory effects [51]. While highly relevant for future investigations, comparing the efficacy of bifrontal vs unilateral treatment efficacy was beyond the scope of the present study. Altogether, we believe that these limiting factors do not invalidate our main claims regarding the existence of executive control deficits associated with MDD and the reduction of this deficit following ECT+DT treatment.

Ultimately the high between-subject variability means that there is no ECT modality that is universally better for all patients, and the results of our study do not intend to promote bifrontal ECT as such a standard.

R1.3. Some studies have found different outcomes of ECT for females compared to males. Possible gender effects on symptom level and cognitive function could be controlled for statistically, or at least mentioned in the Discussion section as a limitation if it is not done.

We thank the reviewer for noticing this. We have now conducted an exploratory analysis to included gender as an additional between-subject factor in the ANOVAs using both frequentist and Bayesian approaches and we did not observe a significant three-way interaction between Gender (male, female), Group (DT, ECT, HC), and Session (Pre, Post) on both RT (F2,90 = 0.22; p =.90; ηp2 = .05; BFincl = 0.21) and ER (F(2,90) F = 0.39, p = 0.68, ηp2 = 0.01; BFincl = 0.31).

Because other reviewers already highlighted the high number of tests we are reporting in the manuscript, we would rather not add another section including this control analysis. We are available to revert this decision in case the reviewer thinks otherwise.

R1.4.  There is a large number of statistical tests being performed on a relatively small sample. Did the authors do any power calculations beforehand? Possibly, the weakest statistically significant results would not withstand a Bonferroni correction. The authors should either perform such correction for multiple testing, or address the absence of such as a limitation.

We thank the reviewer for this comment, which is in line with the comment from another reviewer. Our choice of running multiple nonparametric paired tests, rather than one omnibus parametric test, is due to the violation of some of the assumptions for the ANOVA (e.g., normal distribution). Therefore, we agree that p-values obtained from nonparametric tests (e.g., Wilcoxon signed rank) are likely to be affected by multiple comparisons problem not addressed here. This fact motivated us to approach these paired tests using Bayesian statistics which would provide evidence for the absence of an effect of interest, and the main conclusions of this manuscript are based on this dual (i.e., non parametric & Bayesian) approach.

We have now discussed the multiple comparisons issues and our approach to address it in the Limitation section

R1.5. Related to all of these concerns: There is no Limitation section in the Discussion part. In fact, there is little discussion of the possible shortcomings of this study at all. Therefore, some of the conclusions drawn are premature.

We have now added a limitation section to the manuscript at page 10. The writing added  to the manuscript is also shown below:

A few methodological aspects of our study are worth considering as limitations. First, group assignment was based upon on psychiatrist evaluation of  severity, efficacy of pharmacological intervention, and acute suicidal risk. No patients underwent ECT treatment only, and the determination of ECT+DT or DT for patients’ clinical treatment was unrelated to the scope of the present study. As ECT+DT patients were considered treatment-resistant, it is important to note that while average illness duration of the ECT+DT group was higher in magnitude (though not statistically significant) compared to the DT group, the average HAMD score was significantly lower in the ECT+DT group compared to the DT group. A diagnosis of treatment-resistant depression has to undergo multiple considerations, and hence having a lower HAMD score does not necessarily indicate absence of treatment resistance nor of suicidal ideation. Moreover, the research team only has access to HAMD scores upon enrollment in the study and not at the onset of depression for either group, which prevents the establishment of patients’ clinical status before treatment and the extent of the symptom reduction in drug-refractory patients.

An additional confounding effect can be attributed to our assignment procedure to either the ECT+DT or DT group because ECT treatment was offered only to patients that were treatment-resistant to pharmacological intervention, a characteristic that is not shared with the DT group. Although this assignment procedure is dictated by ethical reasons, it remains important to still consider how the presence of this difference in the two groups might affect our conclusions. While the sample size of the present study may present some limitations to the generalizability of the present study’s findings, our research group has published a fairly high number of articles investigating behavioral, cognitive, and neural markers of attention in a variety of clinical populations e.g., [38, 51]. Based on our previous experience, we believe that these numbers were sufficient to observe our predicted effects on attentional decreases post-treatment, which is why we did not conduct a priori power analysis. Yet, it is important to notice that our cluster analyses are underpowered due to the sample sizes, which is why results must be considered only exploratory.

Variability in the efficacy of ECT treatment according to electrode placement and other methodological differences are key factors relevant to the interpretation of the present findings. The bifrontal electrode placement and ECT session frequency employed in this study was carried out according to the treatment standards of the Second Hospital of Anhui Medical University. We recognize that some studies have shown right unilateral ECT to have greater efficacy than bifrontal ECT [73], and that there is a recent trend attempting to reduce total treatment duration (e.g., ultrabrief unilateral ECT) [74](Shafi et al., 2018). However, the evidence for this difference in efficacy is mixed [75], and it is recognized that differences between individuals (such as head size) may allow bifrontal placement to be better at avoiding brain regions associated with detrimental memory effects [51]. While highly relevant for future investigations, comparing the efficacy of bifrontal vs unilateral treatment efficacy was beyond the scope of the present study. Altogether, we believe that these limiting factors do not invalidate our main claims regarding the existence of executive control deficits associated with MDD and the reduction of this deficit following ECT+DT treatment.

Reviewer 2 Report

Dear Authors

Thank you for the submission

This is an interesting study with a conclusion that ECT can improve executive control deficit but I have significant concerns about the methodology.

It is unclear to me how the 67 participants (later 56 after excluding 11 bipolar cases) were allocated to the Drug Treatment (DT) vs ECT group. It appears that those who were treatment resistant and suicidal got ECT and everyone else got DT? I cannot find any other elaboration or definition of the selection criteria. Even if I could find the criteria this is a clear selection bias and makes all the results difficult to interpret.

If there was a limitation section that discusses this it would at least be helpful. But I am unable to find a limitation section.

I am also unable to find any tables and the supplementary tables and methods are in a Macintosh format unreadable in a Windows computer.

Further the number of subjects (56) is fairly small and further subdivided into 5 more clusters (2 for ECT, 3 for DT). These very small groups further undergo a dizzying variety of statistical tests with no power calculation.

In short I am unable to accept the results or conclusions of this study.

Author Response

We would like to begin by thanking the Editor and the four reviewers for their insightful comments and suggestions. Overall, we believe that information we have now added to the manuscript has significantly improved it, especially regarding how patients were assigned to each group. We hope that the reviewers will find our work to be detailed and accurate. We have put our best effort to add as much information as needed, while at the same time shortening the manuscript.

Below we provide a point by point response to each Reviewers’ points, that are now numbered.  We rest at your availability in case additional information is needed.

Reviewer 2

This is an interesting study with a conclusion that ECT can improve executive control deficit but I have significant concerns about the methodology.

We thank the reviewer for the kind words. We have now modified the manuscript to address those concerns, and we hope that the reviewer will find our changes to suffice. We rest at the Reviewer’s availability in case additional work is needed.

R2.1. It is unclear to me how the 67 participants (later 56 after excluding 11 bipolar cases) were allocated to the Drug Treatment (DT) vs ECT group. It appears that those who were treatment resistant and suicidal got ECT and everyone else got DT? I cannot find any other elaboration or definition of the selection criteria. Even if I could find the criteria this is a clear selection bias and makes all the results difficult to interpret.

If there was a limitation section that discusses this it would at least be helpful. But I am unable to find a limitation section.

We thank the reviewer for raising this point. We have now added additional information about potential limitations regarding the approach we used in group assignment in the Limitation section of the manuscript as shown below:

A few methodological aspects of our study are worth considering as limitations. First, group assignment was based upon on psychiatrist evaluation of  severity, efficacy of pharmacological intervention, and acute suicidal risk. No patients underwent ECT treatment only, and the determination of ECT+DT or DT for patients’ clinical treatment was unrelated to the scope of the present study. As ECT+DT patients were considered treatment-resistant, it is important to note that while average illness duration of the ECT+DT group was higher in magnitude (though not statistically significant) compared to the DT group, the average HAMD score was significantly lower in the ECT+DT group compared to the DT group. A diagnosis of treatment-resistant depression has to undergo multiple considerations, and hence having a lower HAMD score does not necessarily indicate absence of treatment resistance nor of suicidal ideation. Moreover, the research team only has access to HAMD scores upon enrollment in the study and not at the onset of depression for either group, which prevents the establishment of patients’ clinical status before treatment and the extent of the symptom reduction in drug-refractory patients.

R2.2. I am also unable to find any tables and the supplementary tables and methods are in a Macintosh format unreadable in a Windows computer.

We apologize for this glitch. We had submitted the tables to the journal and we are unsure why the submission did not go through as expected. We have now incorporated the tables in the manuscript to ensure access to the content.

R2.3. Further the number of subjects (56) is fairly small and further subdivided into 5 more clusters (2 for ECT, 3 for DT). These very small groups further undergo a dizzying variety of statistical tests with no power calculation.

We agree that the sample size is small, which is why we consider the cluster analyses only exploratory and the main conclusions of the article are not based on this analysis, but on nonparametric and Bayesian statistics (as discussed above). Still, we believe that the cluster analysis, although preliminary, could be helpful for future studies on differential effects of treatment efficacy in MDD, and that is why we would like to report it. Further, tests conducted on the cluster analysis do not affect (retroactively) the results from the nonparametric and Bayesian approaches used. We have now added the following statement in the only sentence that discussed this result in our manuscript:

“It is important to keep in mind that cluster analyses are to be considered exploratory and readers should exercise caution when drawing any interpretation from these results."

Regarding the power analysis calculation, our group has published a fairly high number of articles investigating behavioral, cognitive, and neural markers of attention in a variety of clinical populations (e.g., Schizophrenia - Spagna et al., 2018; 2015 - already cited in the manuscript; MDD - Tian et al., 2016 - already cited in the manuscript; Bai et al., 2017; Wang et al., 2018; Wei et al., 2018; Wang et al., 2018 and others).

Bai, T., Xie, W., Wei, Q., Chen, Y., Mu, J., Tian, Y., & Wang, K. (2017). Electroconvulsive therapy regulates emotional memory bias of depressed patients. Psychiatry Research, 257, 296-302.

Wang, J., Wei, Q., Bai, T., Zhou, X., Sun, H., Becker, B., ... & Kendrick, K. (2017). Electroconvulsive therapy selectively enhanced feedforward connectivity from fusiform face area to amygdala in major depressive disorder. Social cognitive and affective neuroscience, 12(12), 1983-1992.

Wang, J., Wei, Q., Yuan, X., Jiang, X., Xu, J., Zhou, X., ... & Wang, K. (2018). Local functional connectivity density is closely associated with the response of electroconvulsive therapy in major depressive disorder. Journal of affective disorders, 225, 658-664.

Wei, Q., Bai, T., Chen, Y., Ji, G., Hu, X., Xie, W., ... & Tian, Y. (2018). The changes of functional connectivity strength in electroconvulsive therapy for depression: a longitudinal study. Frontiers in neuroscience, 661.

Wang, J., Wei, Q., Wang, L., Zhang, H., Bai, T., Cheng, L., ... & Wang, K. (2018). Functional reorganization of intra‐and internetwork connectivity in major depressive disorder after electroconvulsive therapy. Human Brain Mapping, 39(3), 1403-1411.

We have also added the following text to the limitations section:

An additional confounding effect can be attributed to our assignment procedure to either the ECT+DT or DT group because ECT treatment was offered only to patients that were treatment-resistant to pharmacological intervention, a characteristic that is not shared with the DT group. Although this assignment procedure is dictated by ethical reasons, it remains important to still consider how the presence of this difference in the two groups might affect our conclusions. While the sample size of the present study may present some limitations to the generalizability of the present study’s findings, our research group has published a fairly high number of articles investigating behavioral, cognitive, and neural markers of attention in a variety of clinical populations e.g., [38, 51]. Based on our previous experience, we believe that these numbers were sufficient to observe our predicted effects on attentional decreases post-treatment, which is why we did not conduct a priori power analysis. Yet, it is important to notice that our cluster analyses are underpowered due to the sample sizes, which is why results must be considered only exploratory.

R2.4. In short I am unable to accept the results or conclusions of this study

We hope that the revised version of the manuscript is clearer now, and that the Reviewer would consider it worthy of publication.

Reviewer 3 Report

This article presents an interesting an novel finding of the non-correlation between improvements in depression and in executive control. The results are thoughtfully presented.

Notably, my review copy did not have any tables, so I am unable to assess content in those tables which significantly limits my ability to review the manuscript in a meaningful way.

A few points:

1) were patients randomized to ECT? Lines 138-140 suggest that the more ill patients were assigned to ECT--did any decline this treatment? A participant flow diagram (CONSORT diagram) would be helpful

2) Lines 158-169: this figure does not make it clear to me exactly what is being tested. There are more details in the SI but I think this section needs to be somewhat expanded to make the task clear without reference to the supplement

3) Was there any change to the ECT parameters over the course of treatment or were all patients kept at the same charge dose for each treatment? A table listing more about the ECT (mean number of treatments, seizure duration, mean charge) would be helpful

4) The ECT used for this study (bilateral daily for 3 days than every other day) is much more intensive than used in many other places in the world. Discussion section should discuss how these results may not translate to other forms of ECT (e.g. ultrabrief unilateral)

5) it is interesting to me that the drug treatment group has higher HAMD scores at baseline than the ECT group, despite the ECT patients being more treatment resistant. Do the authors have any thoughts as to why this may be?

6) The cluster analysis 3.3 in the results section is not clear to me. I'm not sure what this adds to the overall paper, and suggest that this be clarified or else omitted. The paper is already fairly long and omitting this could trim it down some

7) 

Author Response

We would like to begin by thanking the Editor and the four reviewers for their insightful comments and suggestions. Overall, we believe that information we have now added to the manuscript has significantly improved it, especially regarding how patients were assigned to each group. We hope that the reviewers will find our work to be detailed and accurate. We have put our best effort to add as much information as needed, while at the same time shortening the manuscript.

Below we provide a point by point response to each Reviewers’ points, that are now numbered.  We rest at your availability in case additional information is needed.

Reviewer 3

This article presents an interesting an novel finding of the non-correlation between improvements in depression and in executive control. The results are thoughtfully presented.

We thank the reviewer for the kind words.

R3.1. Notably, my review copy did not have any tables, so I am unable to assess content in those tables which significantly limits my ability to review the manuscript in a meaningful way.

We apologize for this glitch. We had submitted the tables to the journal and we are unsure why the submission did not go through as expected. We have now incorporated the tables in the manuscript to ensure access to the content.

R3.2. Were patients randomized to ECT? Lines 138-140 suggest that the more ill patients were assigned to ECT--did any decline this treatment? A participant flow diagram (CONSORT diagram) would be helpful

We thank the reviewer for this comment, which was also raised by another reviewer. Participants assignment was already briefly mentioned in the Participants section of the manuscript. We have now added the following text to the Limitations section to further clarify the potential confounding factor of our approach.

A few methodological aspects of our study are worth considering as limitations. First, group assignment was based upon on psychiatrist evaluation of  severity, efficacy of pharmacological intervention, and acute suicidal risk. No patients underwent ECT treatment only, and the determination of ECT+DT or DT for patients’ clinical treatment was unrelated to the scope of the present study. As ECT+DT patients were considered treatment-resistant, it is important to note that while average illness duration of the ECT+DT group was higher in magnitude (though not statistically significant) compared to the DT group, the average HAMD score was significantly lower in the ECT+DT group compared to the DT group. A diagnosis of treatment-resistant depression has to undergo multiple considerations, and hence having a lower HAMD score does not necessarily indicate absence of treatment resistance nor of suicidal ideation. Moreover, the research team only has access to HAMD scores upon enrollment in the study and not at the onset of depression for either group, which prevents the establishment of patients’ clinical status before treatment and the extent of the symptom reduction in drug-refractory patients.

Based on the procedure mentioned above, we believe that a flow-chart would not be informative of the group assignments due to the highly-individualized nature of the treatment received by each patient.

R3.3. Lines 158-169: this figure does not make it clear to me exactly what is being tested. There are more details in the SI but I think this section needs to be somewhat expanded to make the task clear without reference to the supplement

We thank the reviewer for this comment. We have now integrated some of the wording front the Supplementary Material into the main text to provide additional details about the task, as shown below:

“The Lateralized Attention Test – Revised (LANT-R) consists of a simple computerized task requiring the participant to indicate the direction of an arrow, presented at 6 degrees of visual field to the right or to the left of a central fixation point, by performing an up or down button press. The presentation of the target was preceded by one of three cue conditions: 1) double cue; 2) spatial cue; 3) no cue, to manipulate participants attentional orientation to the target.“

R3.4. Was there any change to the ECT parameters over the course of treatment or were all patients kept at the same charge dose for each treatment? A table listing more about the ECT (mean number of treatments, seizure duration, mean charge) would be helpful.

We appreciate the Reviewer pointing at this aspect of our manuscript. Seizure activity was estimated using electroencephalography during ECT, but we do not have a record of mean charge and seizure duration for each individual participant. The ECT procedure detailed in the method section indicates that the average total duration of the treatment was approximately of two weeks and that mean charge was individualized based on patient’s age and induction of seizures, in line with our previous studies (Bai et al., 2017; Wei et al., 2018).

Bai, T., Xie, W., Wei, Q., Chen, Y., Mu, J., Tian, Y., & Wang, K. (2017). Electroconvulsive therapy regulates emotional memory bias of depressed patients. Psychiatry Research, 257, 296-302.

Wei, Q., Bai, T., Chen, Y., Ji, G., Hu, X., Xie, W., ... & Tian, Y. (2018). The changes of functional connectivity strength in electroconvulsive therapy for depression: a longitudinal study. Frontiers in neuroscience, 661.

R3.5. The ECT used for this study (bilateral daily for 3 days than every other day) is much more intensive than used in many other places in the world. Discussion section should discuss how these results may not translate to other forms of ECT (e.g. ultrabrief unilateral)

Variability in the efficacy of ECT treatment according to electrode placement and other methodological differences are key factors relevant to the interpretation of the present findings. The bifrontal electrode placement and ECT session frequency employed in this study was carried out according to the treatment standards of the Second Hospital of Anhui Medical University. We recognize that some studies have shown right unilateral ECT to have greater efficacy than bifrontal ECT [73], and that there is a recent trend attempting to reduce total treatment duration (e.g., ultrabrief unilateral ECT) [74](Shafi et al., 2018). However, the evidence for this difference in efficacy is mixed [75], and it is recognized that differences between individuals (such as head size) may allow bifrontal placement to be better at avoiding brain regions associated with detrimental memory effects [51]. While highly relevant for future investigations, comparing the efficacy of bifrontal vs unilateral treatment efficacy was beyond the scope of the present study. Altogether, we believe that these limiting factors do not invalidate our main claims regarding the existence of executive control deficits associated with MDD and the reduction of this deficit following ECT+DT treatment.

Ultimately the high between-subject variability means that there is no ECT modality that is universally better for all patients, and the results of our study do not intend to promote bifrontal ECT as such a standard.

R3.6. it is interesting to me that the drug treatment group has higher HAMD scores at baseline than the ECT group, despite the ECT patients being more treatment resistant. Do the authors have any thoughts as to why this may be?

This is a very relevant point, and it was raised also by another reviewer of this manuscript. We have now added to the manuscript a limitation section to discuss more in detail our group assignment procedure. Yet, we would like to point out that depression severity and treatment resistance are independent from one another, which can at least partially answer the point raised by the reviewer. We have already mentioned the unusual pattern of lower HAMD scores in the ECT+DT group compared to DT only in page 10 of the manuscript. Therefore, to provide a conclusive answer to this point we would need data that is not available to the research group at the present moment, namely, the HAMD scores of patients from both groups at the onset of depression and presence of acute suicidal ideation. In other words, our psychiatrists proposed a combination of ECT and DT treatment to participants that were treatment resistant and / or presenting suicidal ideation.

Simply put, when diagnosing treatment-resistant depression we have to undergo multiple considerations and having a lower HAMD score does not necessarily indicate absence of treatment resistance.

R3.7.  The cluster analysis 3.3 in the results section is not clear to me. I'm not sure what this adds to the overall paper, and suggest that this be clarified or else omitted. The paper is already fairly long and omitting this could trim it down some

As discussed in response to the comment from another reviewer, we are aware of the exploratory nature of the cluster analyses, and that caution is needed when interpreting these results also in light of the small sample sizes within each cluster. Yet, the main conclusions of the article are not based on these analyses, but on nonparametric and Bayesian statistics (as discussed above). Still, we believe that the cluster analysis, although preliminary, could be helpful for future studies on differential effects of treatment efficacy in MDD, and that is why we would like to report it. Further, tests conducted on the cluster analysis do not affect (retroactively) the results from the nonparametric and Bayesian approaches used. We have now added the following statement in the only sentence that discussed this result in our manuscript:

“It is important to keep in mind that cluster analyses are to be considered exploratory and readers should exercise caution when drawing any interpretation from these results."

We rest at the reviewer’s availability in case they still deem the cluster analyses to be removed.

Reviewer 4 Report

This is an interesting manuscript addressing a relatively poor studied topic: the cognitive symptoms in MDD. As a clinician and researcher I found the work attractive, and I particularly appreciated the attempt by the authors to provide a longitudinal approach by including pre and post measurements. However, I’d like to comment some concerns that have arisen while reviewing this manuscript and propose some aspects to consider in order to further improve this work.

  • Introduction seems a bit long.
  • It would be useful to explain the type of study design (is it prospective?- Please attach the total number of patients that were proposed to participate in the study, and the characteristics comparison of the patients who accepted to participate versus the ones who did not-,or retrospective design using data from the evaluation protocol of your center?).
  • Detail the protocol: who performed the evaluations? Did the subjects had to come to the hospital to do the assessments or were them done in control visits? Were outpatients or inpatients? How did you recruit the healthy control group?
  • How did you assess treatment resistance?
  • The exact time of pre-post test measures of symptoms and of executive control should be explained for each of the three groups (ECT, DT, HC). How many days after finishing ECT was the LANT administered? How did you determine the moment of the post measure in the DT group (response, remission, weeks of treatment?
  • How did you determine seizure adequacy in ECT group?
  • Please include the scores in MMSE or MoCA at pre and post time points.
  • The format of some items in the reference list does not seem consistent with the style used for other references, and there are also different format styles along the text.
  • There are multiple statistical analyses, it would be useful to the readers to explain to which aim of the study refer each one and which are secondary.
  • Were there differences between remitters and non-remitters in LANT results?
  • Do you have an explanation of why the ECT group patients were less severe than the DT ones?
  • Discussion could refer to the differences between objective and subjective measures of cognition and their relationship with patient’s quality of life and functionality.
  • A “methodological issues or limitations” section would be necessary for the present manuscript (e.g. small sample size, absence of quality of life assessment and other cognitive measures)
  • I was not able to find the supplementary results in the files provided.
  • Supplementary tables do not have explanatory legends, the meaning of some abbreviations are not specified and there are some units missing (e.g. course of the disease, dosage)
  • Conclusions of the study are not clearly presented and do not report a summary of the results and their role in the current state of the field.

Author Response

We would like to begin by thanking the Editor and the four reviewers for their insightful comments and suggestions. Overall, we believe that information we have now added to the manuscript has significantly improved it, especially regarding how patients were assigned to each group. We hope that the reviewers will find our work to be detailed and accurate. We have put our best effort to add as much information as needed, while at the same time shortening the manuscript.

Below we provide a point by point response to each Reviewers’ points, that are now numbered.  We rest at your availability in case additional information is needed.

Reviewer 4

This is an interesting manuscript addressing a relatively poor studied topic: the cognitive symptoms in MDD. As a clinician and researcher I found the work attractive, and I particularly appreciated the attempt by the authors to provide a longitudinal approach by including pre and post measurements. However, I’d like to comment on some concerns that have arisen while reviewing this manuscript and propose some aspects to consider in order to further improve this work.

We thank the reviewer for the kind words. We have now modified the manuscript to address those concerns, and we hope that the reviewer will find our changes to suffice. We rest at the Reviewer’s availability in case additional work is needed.

R4.1. Introduction seems a bit long.

We thank the reviewer for providing this comment and we have now attempted to shorten the introduction section. Yet, as is always the case during peer review, while answering the other comments we might have added (rather than took away) some texts (and associated citations). We attempted to do our best to address this comment.

R4.2. It would be useful to explain the type of study design (is it prospective?- Please attach the total number of patients that were proposed to participate in the study, and the characteristics comparison of the patients who accepted to participate versus the ones who did not-,or retrospective design using data from the evaluation protocol of your center?).

This is a relevant point, and it was also raised by another reviewer. Sample size selection for this study was based on our history of publications investigating behavioral, cognitive, and neural markers of attention in a variety of clinical populations (e.g., Schizophrenia - Spagna et al., 2018; 2015 - already cited in the manuscript; MDD - Tian et al., 2016 - already cited in the manuscript; Bai et al., 2017; Wang et al., 2018; Wei et al., 2018; Wang et al., 2018 and others).

Bai, T., Xie, W., Wei, Q., Chen, Y., Mu, J., Tian, Y., & Wang, K. (2017). Electroconvulsive therapy regulates emotional memory bias of depressed patients. Psychiatry Research, 257, 296-302.

Wang, J., Wei, Q., Bai, T., Zhou, X., Sun, H., Becker, B., ... & Kendrick, K. (2017). Electroconvulsive therapy selectively enhanced feedforward connectivity from fusiform face area to amygdala in major depressive disorder. Social cognitive and affective neuroscience, 12(12), 1983-1992.

Wang, J., Wei, Q., Yuan, X., Jiang, X., Xu, J., Zhou, X., ... & Wang, K. (2018). Local functional connectivity density is closely associated with the response of electroconvulsive therapy in major depressive disorder. Journal of affective disorders, 225, 658-664.

Wei, Q., Bai, T., Chen, Y., Ji, G., Hu, X., Xie, W., ... & Tian, Y. (2018). The changes of functional connectivity strength in electroconvulsive therapy for depression: a longitudinal study. Frontiers in neuroscience, 661.

Wang, J., Wei, Q., Wang, L., Zhang, H., Bai, T., Cheng, L., ... & Wang, K. (2018). Functional reorganization of intra‐and internetwork connectivity in major depressive disorder after electroconvulsive therapy. Human Brain Mapping, 39(3), 1403-1411.

In our manuscript, sample size was comparable to our previous studies (e.g., HC in Spagna et al., 2015 = 40; HC in Spagna et al,, 2018 = 40; HC in Tian et al., 2016 = 30; HC in our manuscript = 40; MDD in Tian et al., 2016 pre-post treatment effect study = 24; MDD in Wang et al., 2018 = 23; MDD in our pre-post treatment manuscript = 23 for ECT+DT group and 33 in DT group). Based on our previous experience, we believe that these numbers were sufficient to observe our predicted effects on attentional decreases post-treatment. We therefore took the liberty to avoid conducting a power analysis a priori.

R4.3. Detail the protocol: who performed the evaluations? Did the subjects had to come to the hospital to do the assessments or were them done in control visits? Were outpatients or inpatients? How did you recruit the healthy control group?

Patients were recruited from the Anhui Mental Health Center. These patients received a recommendation to undergo ECT because of their resistance to drug therapies and / or presence of acute suicidal ideation and tendency. They were outpatients, and we have now specified this aspect in the Participant section of the Method. To reiterate, we did not assign participants to any treatment. Rather, participants that were assigned to either ECT+DT or DT only by independent psychiatrists were recruited for this study. The healthy control group was recruited through advertisement in the local area of the Mental Health Center.

We have now revised the manuscript to clarify this aspect, as shown below.

Patients with treatment-resistant depression and/or suicidal ideation who were prescribed ECT were recruited to participate in our study as part of the ECT+DT group (n = 23; 4 males, 19 females). Thirty-three patients entered our DT group (n = 33; 8 males, 25 females).

R4.4. How did you assess treatment resistance?

Lack of remission (i.e., DT resistance) was defined as an HAMD score above 7 after DT treatment.

R4.5. The exact time of pre-post test measures of symptoms and of executive control should be explained for each of the three groups (ECT, DT, HC). How many days after finishing ECT was the LANT administered? How did you determine the moment of the post measure in the DT group (response, remission, weeks of treatment?

We thank the reviewer for pointing out the absence of these details in our manuscript.

Please, see below for the information about the timing of pre-post measures of symptoms (page 3 of the manuscript) in the ECT group.

The 17-item Hamilton Rating Scale for Depression (HAMD) [43] was used to measure the severity of clinical symptoms and was administered 12-24 hours before the beginning of the first testing session (pre-test) and between 24-72 hours from the last testing session (post-test).

We have now added to the manuscript the following sentence, specifying the timing of the LANT administration.

The time between pre and post assessments for the DT group was 28 ± 68 days on average. Both the ECT+DT group and the DT group completed the LANT right after completing the pre and post HAMD evaluations.

For the DT group, the average time between pre and post assessment was 28 days ± 68 days.

We have now added to the manuscript the following sentence, specifying the timing of the LANT administration.

The time between pre and post assessments for the DT group was 28 ± 68 days on average.

For the HC group, the average time between pre and post assessment was 21 days ± 4 days,

We have now added to the manuscript the following sentence, specifying the timing of the LANT administration.

The time between pre and post assessments for the DT group was 21 ± 4 days on average.

R4.6. How did you determine seizure adequacy in ECT group?

Seizure activity was estimated using electroencephalography during ECT. The ECT procedure detailed in the method section indicates that the average total duration of the treatment was approximately of two weeks and that mean charge was individualized based on patient’s age and induction of seizures, in line with our previous studies (Bai et al., 2017; Wei et al., 2018). During administration of ECT, patients were anesthetized with propofol and muscle relaxants were administered.

We have now added the following writing to the manuscript:

Seizure activity was monitored and estimated using electroencephalography, performed concurrently with ECT.

Bai, T., Xie, W., Wei, Q., Chen, Y., Mu, J., Tian, Y., & Wang, K. (2017). Electroconvulsive therapy regulates emotional memory bias of depressed patients. Psychiatry Research, 257, 296-302.

Wei, Q., Bai, T., Chen, Y., Ji, G., Hu, X., Xie, W., ... & Tian, Y. (2018). The changes of functional connectivity strength in electroconvulsive therapy for depression: a longitudinal study. Frontiers in neuroscience, 661.

R4.7. Please include the scores in MMSE or MoCA at pre and post time points.

Due to the outpatient nature of the study, we are not able to retrieve the individual scores at the MMSE. Yet, we have specified our exclusion criteria stating that participants with a score of 24 or lower to the MMSE were excluded.

Exclusion criteria included the following: history of brain tumor, stroke, or other neurological diseases that could disrupt brain function; presence of psychotic or organic mental disorders; diagnosis of bipolar I disorder; current alcohol or drug dependence; diagnosis of borderline or antisocial personality disorder; current treatment with other psychotropic medications; current or recent pregnancy; a score below 24 at the Mini Mental State Examination; less than 5 years of schooling.

R4.8. The format of some items in the reference list does not seem consistent with the style used for other references, and there are also different format styles along the text.

We thank the reviewer for this comment and apologize for the oversight; reference list and in-text citations were created using Zotero and in the format recommended by the Journal. We have now deleted the in-text citations that were not numbered (e.g., in the Discussion), and further updated the reference list to reflect the style used by the journal to the best of our knowledge.

R4.9. There are multiple statistical analyses, it would be useful to the readers to explain to which aim of the study refer each one and which are secondary.

We thank the reviewer for this insightful comment, we have now specified that the Cluster Analyses and Correlation analyses were exploratory in nature and only secondary to the main scope of the article:

3.3. Exploratory Cluster Analysis conducted separately for the ECT and DT groups on RT CE

3.5. Exploratory Correlations analyses between executive control function, clinical symptoms, and illness duration

Further, in response to a previous reviewer we have modified the discussion section to further specify that interpretation of cluster analyses results needs to be drawn with caution, and added a limitation section to the Discussion.

R4.10. Were there differences between remitters and non-remitters in LANT results?

R4.11. Do you have an explanation of why the ECT group patients were less severe than the DT ones?

Our answer here can only be speculative. The ECT group on average showed longer illness duration (i.e., the time passed after first diagnosis). Therefore, it may be that treatment was not entirely ineffective in this group of patients, or that this group of participants had developed some sort of psychological mechanisms to deal with some of the symptoms over time (and therefore report lower scores in the HAMD). This interpretation goes hand in hand with the fact that patients in the ECT group had drug-resistant depression. We have discussed this aspect of the group assignment as follows:

A few methodological aspects of our study are worth considering as limitations. First, group assignment was based upon on psychiatrist evaluation of  severity, efficacy of pharmacological intervention, and acute suicidal risk. No patients underwent ECT treatment only, and the determination of ECT+DT or DT for patients’ clinical treatment was unrelated to the scope of the present study. As ECT+DT patients were considered treatment-resistant, it is important to note that while average illness duration of the ECT+DT group was higher in magnitude (though not statistically significant) compared to the DT group, the average HAMD score was significantly lower in the ECT+DT group compared to the DT group. A diagnosis of treatment-resistant depression has to undergo multiple considerations, and hence having a lower HAMD score does not necessarily indicate absence of treatment resistance nor of suicidal ideation. Moreover, the research team only has access to HAMD scores upon enrollment in the study and not at the onset of depression for either group, which prevents the establishment of patients’ clinical status before treatment and the extent of the symptom reduction in drug-refractory patients.

R4.12. Discussion could refer to the differences between objective and subjective measures of cognition and their relationship with patient’s quality of life and functionality.

We agree with the reviewer that the Discussion could be expanded to include this point. However, we find ourselves a bit puzzled by the simultaneous request from other reviewers to shorten the length of the manuscript. Yet, due to the extensive reviews performed, our manuscript ended up being slightly longer (and luckily more precise). We hope that the reviewer will give us a pass on this comment, so that we can balance the comments and requests from the four reviewers.

R4.13. A “methodological issues or limitations” section would be necessary for the present manuscript (e.g. small sample size, absence of quality of life assessment and other cognitive measures)

We thank the reviewer and have now added a limitation section to the manuscript, as shown below:

A few methodological aspects of our study are worth considering as limitations. First, group assignment was based upon on psychiatrist evaluation of  severity, efficacy of pharmacological intervention, and acute suicidal risk. No patients underwent ECT treatment only, and the determination of ECT+DT or DT for patients’ clinical treatment was unrelated to the scope of the present study. As ECT+DT patients were considered treatment-resistant, it is important to note that while average illness duration of the ECT+DT group was higher in magnitude (though not statistically significant) compared to the DT group, the average HAMD score was significantly lower in the ECT+DT group compared to the DT group. A diagnosis of treatment-resistant depression has to undergo multiple considerations, and hence having a lower HAMD score does not necessarily indicate absence of treatment resistance nor of suicidal ideation. Moreover, the research team only has access to HAMD scores upon enrollment in the study and not at the onset of depression for either group, which prevents the establishment of patients’ clinical status before treatment and the extent of the symptom reduction in drug-refractory patients.

An additional confounding effect can be attributed to our assignment procedure to either the ECT+DT or DT group because ECT treatment was offered only to patients that were treatment-resistant to pharmacological intervention, a characteristic that is not shared with the DT group. Although this assignment procedure is dictated by ethical reasons, it remains important to still consider how the presence of this difference in the two groups might affect our conclusions. While the sample size of the present study may present some limitations to the generalizability of the present study’s findings, our research group has published a fairly high number of articles investigating behavioral, cognitive, and neural markers of attention in a variety of clinical populations e.g., [38, 51]. Based on our previous experience, we believe that these numbers were sufficient to observe our predicted effects on attentional decreases post-treatment, which is why we did not conduct a priori power analysis. Yet, it is important to notice that our cluster analyses are underpowered due to the sample sizes, which is why results must be considered only exploratory.

Variability in the efficacy of ECT treatment according to electrode placement and other methodological differences are key factors relevant to the interpretation of the present findings. The bifrontal electrode placement and ECT session frequency employed in this study was carried out according to the treatment standards of the Second Hospital of Anhui Medical University. We recognize that some studies have shown right unilateral ECT to have greater efficacy than bifrontal ECT [73], and that there is a recent trend attempting to reduce total treatment duration (e.g., ultrabrief unilateral ECT) [74](Shafi et al., 2018). However, the evidence for this difference in efficacy is mixed [75], and it is recognized that differences between individuals (such as head size) may allow bifrontal placement to be better at avoiding brain regions associated with detrimental memory effects [51]. While highly relevant for future investigations, comparing the efficacy of bifrontal vs unilateral treatment efficacy was beyond the scope of the present study. Altogether, we believe that these limiting factors do not invalidate our main claims regarding the existence of executive control deficits associated with MDD and the reduction of this deficit following ECT+DT treatment.

R4.14. I was not able to find the supplementary results in the files provided.

We apologize for this glitch. We had submitted the file to the journal and we are unsure why the submission did not go through as expected. We will reupload the file results and ensure access to the content.

R4.15. Supplementary tables do not have explanatory legends, the meaning of some abbreviations are not specified and there are some units missing (e.g. course of the disease, dosage)

We thank the reviewer for this point. We have now added notes to the tables where acronyms are listed.

R4.16. Conclusions of the study are not clearly presented and do not report a summary of the results and their role in the current state of the field.

We would like to ask the reviewer to be more specific about which parts are not clearly presented, so that we can work on those parts (or is it all of it?) and ensure that the manuscript reaches its highest potential. In the meantime, we hope that the extensive reviews and changes we provided to the manuscript will have helped in this sense.

We rest at your availability in case additional info is needed.

Round 2

Reviewer 1 Report

The authors have done a good job revising and improving the manuscript, and I now recommend it for publication. 

Reviewer 2 Report

Thank you for addressing my concerns. With the added limitations I am ok with the paper.